# Psoriatic arthritis screening: A systematic literature review and experts' recommendations

Ana Urruticoechea-Arana[1], Diego Benavent[2], Fernando León[3], Raquel Almodovar[4], Isabel Belinchón[5], Pablo de la Cueva[6], Cristina Fernández-Carballido[7], Estíbaliz Loza[8], Jordi Gratacós[9]*, Cribado Working Group[¶]

1 Rheumatology Department, Hospital Can Misses, Ibiza, Spain, 2 Rheumatology Department, Hospital Universitario La Paz, IdiPAZ, Madrid, Spain, 3 Primary Care, San Juan de la Cruz Health Center, Pozuelo de Alarcón, Madrid, Spain, 4 Rheumatology Department, Hospital Universitario Fundación Alcorcón, Madrid, Spain, 5 Department of Dermatology, Hospital General Universitario de Alicante-ISABIAL-UMH, Alicante, Spain, 6 Dermatology Department, Hospital Universitario Infanta Leonor, Madrid, Spain, 7 Rheumatology Department, Hospital Universitario de Elda, Alicante, Spain, 8 Instituto de Salud Musculoesquelética, InMusc, Madrid, Spain, 9 Rheumatology Department, Hospital Universitari Parc Taulí, Sabadell, Spain

¶ Membership of the Cribado Working Group is listed in the Acknowledgments.
* jgratacosmas@gmail.com

**Data Availability Statement:** All relevant data are within the paper and its supporting Information files.

## Abstract

### Objective

To analyze the performance of psoriatic arthritis (PsA) screening tools, examine their implementation in daily practice, and reach a consensus about the best screening tool for implementation in daily practice in different medical settings.

### Methods

A systematic literature review (SLR), structured telephone interviews to hospitals, and a multidisciplinary nominal group meeting were all conducted. The SLR employed sensitive search strategies using Medline, Embase, and the Cochrane Library up to January 2020. Two reviewers independently selected articles that reported data on PsA screening tools and that included sufficient data to at least calculate the sensitivity and specificity of those tools (*e.g.*, questionnaires, algorithms, specific questions, and biomarkers). The hospital interviews collected data regarding the process of suspected PsA diagnosis and referral to rheumatology, the implementation of PsA screening tools, and barriers and facilitators to implementation of those tools. In the nominal group meeting, a multidisciplinary team of experts discussed all these data and subsequently recommended a screening tool for implementation.

### Results

The SLR included 41 moderate-quality studies that analyzed 14 PsA screening tools, most of which were questionnaire-based tools. All of these studies reported a moderate-good performance but presented different characteristics regarding the time to completion or the

**Funding:** This study was funded by an independent and unrestricted grant from Pfizer (Grant Number 53038823) awarded to JG and AUA. The funders had no role in study design, data collection and analysis, decision to publish, or preparation of the manuscript.

**Competing interests:** The authors have read the journal's policy and have the following competing interests: IB acted as a consultant and/or speaker for and/or participated in clinical trials sponsored by companies that manufacture drugs used for the treatment of psoriasis, including Janssen Pharmaceuticals Inc, Almirall SA, Lilly, AbbVie, Novartis, Celgene, Amgen, Leo-Pharma, Pfizer-Wyeth, UCB, and MSD; EL has received research grants from Roche, AbbVie, Novartis, Amgen, Leo-Pharma, Pfizer-Wyeth, UCB, Astellas, BMS, Sanofi and MSD but these research grants received by EL are not associated with this study; DB received grants/speaker/research supports from Roche, Novartis and Abbvie but these grants and support received by DB are not associated with this study; PDC acted as consultant, advisory board member, honorary for speaking and participation in clinical trials with the following pharmaceutical companies: Abbvie, Almirall, Astellas, Biogen, Boehringer, Celgene, Janssen, LEO Pharma, Lilly, MSD, Novartis, Pfizer, Roche, Sanofi, UCB, and did not receive any funding, including in the form of salary, for this study from mentioned entities. FL has received honoraria from Novartis. This does not alter our adherence to PLOS ONE policies on sharing data and materials. There are no patents, products in development or marketed products associated with this research to declare.

number and type of items or questions. The implementation of screening tools was low (30.5%). The experts ultimately recommended regular use of a PsA screening tool, preferably the PURE-4 questionnaire.

## Conclusions

The implementation of PsA screening tools like the PURE-4 questionnaire in daily practice likely improves the prognosis of PsA patients.

## Introduction

Psoriatic arthritis (PsA) is a prevalent chronic inflammatory disease that is associated with joint destruction and disability [1, 2]. According to previous studies, up to 30% of patients with psoriasis will develop PsA during the disease course, mostly after the onset of psoriasis and usually within 10 years of their skin disease's first manifestations [3]. PsA is a heterogeneous and complex disease that has several patterns of joint involvement, including axial and peripheral disease, and other extra-articular manifestations like enthesitis, dactylitis, and uveitis [2].

Currently, undiagnosed PsA is common in patients with psoriasis. Several studies and meta-analyses reveal that between 5% and 15.5% of patients with psoriasis may have undiagnosed PsA [4, 5]. Interestingly, it has also been shown that even a six-month delay in the diagnosis of PsA from symptom onset is associated with structural damage and worse long-term physical function [6]. Accordingly, several national and international consensus documents, as well as other projects, recommend a multidisciplinary approach that includes primary care physicians, dermatologists and rheumatologists, in order to establish effective strategies for early and accurate PsA detection [7, 8].

Given this context, different variables have been identified to predict whether a person with psoriasis may develop PsA, including clinical characteristics of psoriasis such as the severity of psoriasis or the presence of certain features of psoriasis (*e.g.*, scalp lesions, nail disease, and intergluteal/perianal psoriasis), the presence of soluble biomarkers, like highly sensitive C-reactive protein or certain susceptibility genes [9–11]. However, their prediction role is or remains unclear [10, 12].

On the other hand, different questionnaires and strategies can be used to detect PsA in patients with psoriasis. Several simple and validated screening tests have been proposed, including the Psoriatic Arthritis Screening and Evaluation (PASE) tool [13], the Psoriasis Epidemiology Screening Tool (PEST) [14], the Toronto Psoriatic Arthritis Screen (ToPAS) [15], and the Early Arthritis for Psoriatic Patients (EARP) questionnaire [16]. More recently, the Psoriatic arthritis UnclutteRed screening Evaluation (PURE-4) questionnaire [17, 18]. However, the heterogeneity, paucity of data regarding feasibility and applicability in clinical settings, and lack of consensus regarding which tool is best all hinder the widespread use of such screening tests [19].

Based on the details outlined above, we decided to perform a systematic literature review (SLR) to assess the performance of available PsA screening tools. We also conducted telephone interviews with rheumatology and dermatology departments to analyze the level of implementation of these tools and to identify barriers and facilitators to their implementation. Finally, we assembled a multidisciplinary nominal group to discuss which screening tool is most appropriate for implementation and to identify further actions.

## Methods

### Design

This project was conducted using a pragmatic approach. First, we conducted an SLR of existing evidence in order to assess the performance of screening tools. Second, through structured telephone interviews with rheumatology and dermatology departments, we analyzed the implementation level of screening tools, as well as their related barriers and facilitators. Finally, a multidisciplinary nominal group meeting was held to discuss the results of the previous steps. This project was reviewed and approved by the ethical oversight committee of the Parc Taulí Hospital Universitari. This study was also conducted in accordance with Good Clinical Practice guidelines and the current version of the revised World Medical Association's Declaration of Helsinki.

Participants consent was informed and obtained verbally. Before starting the telephone interview, participants were informed verbally by the interviewer about the objectives of the project, and that the results of the project were going be published in a scientific journal. It was also explained to the participants that the project would not collect any personal data. Then, all participants were asked to authorize and voluntary consent their participation in the project. If they accepted the conditions, the telephone interview continued, and if they did not, the telephone interview did not take place. To document the verbal consent, the interviews were recorded.

### Systematic review of the literature

The coordinators of the study generated a review protocol that followed the structure of the Cochrane Collaboration. This protocol included the patient-intervention-comparison (PICO) question and selection criteria, as well as procedures and key terms for the design of the search strategies.

Studies were identified using sensitive search strategies in the main medical databases. For this purpose, an expert librarian designed and checked the search strategies. The strategy combined disease- and screening tool-related terms with a controlled vocabulary for specific MeSH headings and additional keywords. This included keywords like "psoriatic arthritis" and "screening" (see S1–S3 Tables in the S1 File). The following databases were screened: Medline (PubMed) and Embase (Embase.com) from 1961 to January 2020, and the Cochrane Library until January 2020. All the retrieved references were managed using Endnote X5 (Thomson Reuters). Due to the number of retrieved articles from the main databases, we decided not to further investigate scientific meeting abstracts or other non-peer reviewed sources. Finally, a hand search was performed to review the references of the included studies.

Studies were included if they met the following pre-established inclusion criteria. Articles had to report on a PsA screening tool and had to include sufficient data to enable us to at least calculate the tool's sensitivity and specificity. This included questionnaires, algorithms, the application of specific questions, and biomarkers, among others. There were no restrictions regarding the type of screening tool or setting (primary or secondary care). SLRs, randomized controlled trials (RCTs), and observational studies in English, French, or Spanish were all considered.

Screening of the studies, data collection, and analysis were performed by two reviewers (EL and DB). Both reviewers independently screened the titles and abstracts of the retrieved articles, considering the selection criteria. In the event of discrepancy, the reviewers discussed the articles in an attempt to reach an agreement. When consensus was not reached, a third reviewer (LC) was asked to resolve the issue. Articles from the previous selection process were then read in detail, which ultimately resulted in establishing a list of included studies. Data

collection was also doubled by article and was independent. Similarly, when discrepancies were not resolved, the third reviewer (LC) took the final decision. The QUADAS-2 score was used to grade the quality of the RCTs [20].

Finally, evidence tables were produced that described the main characteristics of the included studies. Descriptive Results were expressed as either a number and percentage (%) for categorical tests and mean and standard deviation (SD), or median and interquartile range (p25–p75) for normal and no normal continuous variables, respectively. Information regarding diagnostic efficacy was collected through the validation dimensions provided in the articles included in the SLR: internal consistency (Cronbach's alpha), intra- and interobserver reliability (intraclass correlation coefficient in quantitative tests and kappa in categorical tests), criterion validity (sensitivity, specificity, predictive values and likelihood ratios), and overall value of quantitative tests (ROC curve and AUC, along with their confidence interval). Meta-analysis was only planned for cases of homogeneity.

### Hospital telephone interviews

We decided to invite a minimum of 60 hospitals, using the following selection criteria. Hospitals had to be part of the National Health System, with the availability of rheumatology and dermatology departments, geographical representativeness (centers all over the country), and representativeness regarding the type of hospitals (*e.g.*, general hospitals, county hospitals). The National Hospital Database was used to select the centers. The people responsible for the rheumatology and dermatology departments were contacted for interviews.

Subsequently, a structured telephone interview was conducted in those hospitals that agreed to participate. A trained researcher asked several questions regarding the following main topics: the process of suspected PsA diagnosis (in dermatology and primary care) and referral to rheumatology; the use of PsA screening tools, strategies, and pre-established criteria; and barriers and facilitators to implementation of PsA screening tools.

The structured telephone survey also collected data regarding the hospital's features and its rheumatology and dermatology departments (*e.g.*, attended population, number of health professionals in the department, presence of a multidisciplinary care model).

### Multidisciplinary nominal group meeting

Finally, a multidisciplinary nominal group meeting was held via video conference. This group comprised five rheumatologists, two dermatologists, and one primary care physician, who were selected according to the following criteria: 1) Rheumatologist, dermatologist or primary care physician; 2) Specialized in psoriatic disease with demonstrated clinical experience; 3) Clinical experience ≥8 years and/or ≥5 publications; 4) Participation in multidisciplinary projects in psoriasis and PsA; 5) Members of national and international medical societies. In the selection process a balanced territorial representation of Spain was considered. The number of rheumatologists was a bit higher compared with other health professionals because they have been specifically working in this field for a long time.

The group discussed the results of the SLR and telephonic interviews, and they then agreed on the best screening tool, which they recommend for implementation.

## Results

### Systematic review of the literature

The search strategies retrieved 3,084 citations (including 429 duplicates). A flowchart summarizing the search results is presented in Fig 1. After the first selection process, 2,594 citations

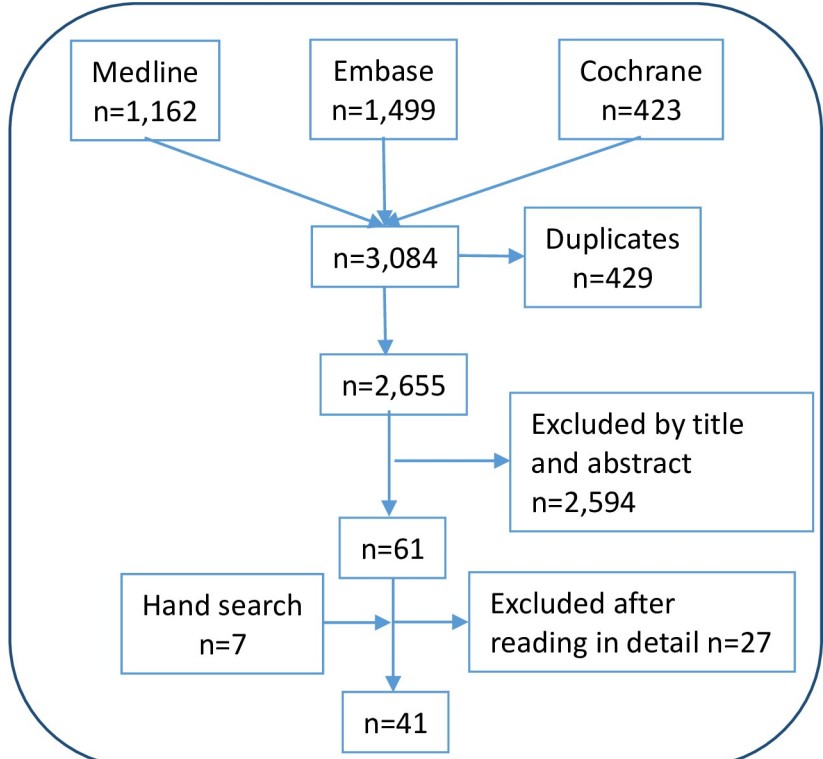

**Fig 1. Studies flow-chart.**

were rejected and 61 were selected for detailed review. Subsequently, 27 citations were excluded [19, 21–46], mainly because they did not provide data on screening tools' performance (see S4 Table in the S1 File). Finally, 41 studies were included (including seven from the secondary search) [10, 13–17, 41, 47–81].

The main characteristics of the included studies are outlined in Table 1 and the results are presented in Table 2. Most of the included studies were validation studies [13–17, 47, 50, 54, 58, 64, 68, 77, 78], cross-cultural validation studies [49, 51, 52, 54, 59, 61–63, 71–73], or diagnostic performance studies [10, 53, 55–57, 60, 65, 67, 69, 70, 74, 75, 80, 81]. We also found other designs, including case-control studies [48], clinical trials [66], and prospective observational designs [79]. The oldest article was published in 2007 [13], while the most recent one was published in 2019 [47, 49, 72, 73]. The quality, according to the QUADAS-2 scale, was variable but was moderate in most cases (see S5 Table in the S1 File for more information). However, when the QUADAS-2 scale was evaluated, the quality of the articles tended to decrease as the risk of bias was high or unclear in many articles, especially in terms of patient selection. Similarly, it must be noted that cross-cultural validations do not describe all the necessary requirements to be considered adequate, so we must be very careful with their results.

We found great variability in the features of the patient samples. Some patients were recruited in primary care, while others were recruited in outpatient dermatology or rheumatology consultations (e.g., centers of excellence, specific consultations, multidisciplinary units, and regular consultations). The number of included patients varied from 54 [68] to 1,225 [74]. The percentage of men was slightly higher than women, and most patients were aged between 40 and 55 years. The duration of psoriasis also varied, with most patients presenting long-standing disease ranging between five [68] and 31 years [62]. Although baseline treatments

**Table 1. Evidence table.** Main characteristics of the included studies.

| # | Study | Population | Intervention/s | Gold standard | Measures |
|---|-------|-----------|----------------|---------------|----------|
| 1 | **Audureau_2018** [17], validity study, France | • n = 137 Pso, median age 43 yry, median Pso duration 12 yr, 87.6% plaque Pso, 16.3% severe Pso | • PURE-4 scale | • CASPAR criteria | • Se; Sp<br>• PV+; PV-<br>• AUC |
| 2 | **Chandran_2011** [53], cross-sectional, Canada | • n = 134 patients from a PsA clinic; n = 123 patients from a psoriasis clinic; n = 118 patients from a general dermatology clinic had PsA; n = 135 patients from a general rheumatology clinic; n = 178 patients from a family medicine clinic | • ToPAST | • 4 Rheumatologists consensus | • Se; Sp<br>• PV+; PV-<br>• AUC |
| 3 | **Chimenti_2019** [47], validity study, Italy | • n = 239 Pso and musculoskeletal pain<br>• 54% women; mean age 51.2±13.4 yr from<br>• dermatological/rheumatological centres<br>• 120/239 (50.2%) PsA | • PsA-Disk (visual instrument for musculoskeletal symptoms)<br>• PEST (for concurrent validity) | • CASPAR criteria | • Internal consistency: α Cronbach<br>• Test-retest: ICC<br>• Concurrent V: kappa<br>• Construct: known groups |
| 4 | **Chiowchanwisawakit_2016** [54], transcultural validation and validity study, Thailand | Pso clinic at university hospital<br>• n = 159: Ps (34) vs PsA (125); median age 47.4 vs 46.3; median Pso duration (yr): 9.0 vs 12.0 | • Thai version of PEST<br>• Thai version of EARP<br>• Development new tool: SiPAT | • CASPAR criteria | • Se; Sp<br>• PV+; PV+<br>• AUC<br>• LR |
| 5 | **Coates_2013** [55], cross-sectional study, CONTEST study, UK | • Hospital based-sample<br>• n = 195 Pso; 47 PsA (24.1%); 49.7% women; mean age 47 yr; Pso mean duration 22.5 yr | • PEST<br>• PASE<br>• ToPAS | • CASPAR criteria | • Se; Sp<br>• AUC |
| 6 | **Coates_2014** [58], validation study, international | • n = 195<br>Validation cohorts:<br>• Dublin: 100 Pso; 29 PsA<br>• Utha: 145 Pso; 80 PsA | Different methods:<br>• Most discriminatory questions (CONTEST)<br>• Weighted questions (CONTESTw)<br>• Joint manikin (CONTESTjt)<br>• CART analysis (CONTESTtree) | • CASPAR criteria | • Cut-off point and Se<br>• AUC |
| 7 | **Coates_2016** [56], cross-sectional, UK | Primary care setting<br>169 Pso; 17 (10.1%) PsA<br>• Median age: 61.0 vs 52 yr<br>• Pso duration: 28 vs 30 yr<br>• Plaque pso: 79.2% vs 83.3% | • CONTEST<br>• PEST<br>• PASE<br>• ToPAS | • Diagnosis by rheumatologist | • Se; Sp<br>• AUC |
| 8 | **Coates_2018** [57], cross-sectional, UK | Secondary dermatology clinics<br>• n = 159; 27 (17%) PsA<br>PsA: older, more severe skin and nail disease | • CONTEST<br>• PEST | • Diagnosis by rheumatologist | • Se; Sp<br>• AUC |
| 9 | **Costa_2018** [59], cross-cultural validation, Portugal | Dermatology outpatient clinics<br>• n = 465 Pso; 158 PsA (33.9%)<br>• Mean age: 48.8 yr; -Women 50%<br>• Mean duration Pso: 15.5 yr | • PASE-P (Portuguese version) | • CASPAR criteria | • Reliability:<br>• Internal consistency: α Cronbach<br>• Test-retest: ICC<br>• Validity<br>• Discriminant: ρ Spearman<br>• Known groups: Pso vs PsA<br>• Construct: Factor analysis |
| 10 | **Cretu_2018** [48], case-control study, Canada | Serum biomarkers to differentiate Pso–PsA<br>• Pso: n = 100; mean age/Pso duration: 51/22.9<br>• PsA: n = 100; mean age/Pso duration: 49/20<br>• Control n = 100; mean age: 35 | • Mac-2-binding protein (M2BP)<br>• CD5-like protein (CD5L)<br>• Myeloperoxidase (MPO)<br>• Integrin b5 (ITGβ5)<br>• Matrix metalloproteinase 3<br>• CRP level | • CASPAR criteria | • Association of each marker with the 3 study groups: polychotomous logistic regression<br>• Accuracy: ROC curve/AUC<br>• Correlation biomarkers: ρ Spearman |

(*Continued*)

**Table 1.** (Continued)

| # | Study | Population | Intervention/s | Gold standard | Measures |
|---|-------|-----------|----------------|---------------|----------|
| 11 | **Dominguez_2009** [60], cross-sectional, USA | Tertiary care center <br> • n = 190 Pso; 37 (19.5%) PsA <br> 57.8% male; 72.6% Caucasian | • PASE | • Moll & Wright criteria | • Se; Sp <br> • AUC <br> • Test-retest: ρ Pearson, ICC <br> • Sensitivity to change: paired t test |
| 12 | **Duruoz_2018** [61], cross-cultural validation, Turkey | Hospital outpatient clinics. Volunteers <br> • n = 150; mean age 41.1; 58% female, 43 PsA; 46 Pso; 41 physical medicine; 20 rheumatology | • ToPAS 2 | • CASPAR criteria | • Se; Sp <br> • AUC |
| 13 | **Fernández_Ávila_2017** [62], cross-cultural validation, Colombia | n = 108; 36 (33%) PsA; 60% male; mean age: 51.2 yr; Pso duration 31.5 yr; PsA duration 6.0 yr | • ToPAS (Spanish version; Colombia) | • CASPAR criteria | • Se; Sp; <br> • PPV; NPV <br> • FA <br> • Internal consistency: KR20 <br> • Test-retest: ρ Pearson |
| 14 | **Ferreyra Garrott_2013** [63], cross-cultural validation, Argentina | Hospital derma and rheumatology outpatient clinics <br> • n = 111; mean age = 56.9 yr (23 Pso; 25 PsA; 22 Pso + OA; 41 OA) | • PASE (Spanish version; Argentina) | • CASPAR criteria | • Se; Sp <br> • AUC |
| 15 | **Garg_2015** [64], validity study, USA | Center of Excellence for Pso and PsA <br> • n = 517 Pso; 117 (22.6%) PsA <br> • mean age: 46.3 yr; female = 55.5% <br> • mean duration of Pso = 17.9 yr | • CEPPA | • Diagnosis by rheumatologist | • Se; Sp <br> • PV+; PV- <br> • AUC <br> • LR |
| 16 | **Gladman_2009** [15], validation study, Canada | 5 groups of patients <br> • University PsA Clinic (n = 134 PsA) <br> • Psoriasis Centre (n = 123 Pso; 30 PsA (24.4%) <br> • Dermatology clinic (n = 118; 2(1.7%) PsA) <br> • Rheumatology clinic (n = 135; 0 PsA) <br> • Family medicine clinics (n = 178; 3 (1.7%) PsA) | • ToPAS | • CASPAR criteria | • Stepwise logistic regression to identify the most discriminative questions between Pso and PsA <br> • Logistic model fitted to 3 relevant domains for PsA skin, joints, nails <br> • Weighting of each domains <br> • ROC curve for each model |
| 17 | **Haddad_2019** [49], cross-cultural validation (Hebrew) | • Dermatology and combined rheumatology-dermatology clinics: n = 93 <br> • Primary care dermatology clinic: n = 115 <br> • PsA n = 108 (51.9%) | • PASE <br> • PEST <br> • ToPAS <br> • EARP <br> • CONTEST | • CASPAR criteria | • Se; Sp <br> • PV+; PV- <br> • AUC |
| 18 | **Härle_2017** [65], cross-sectional, Germany | • 59 dermatology units <br> • n = 1.002; mean age 49.4 yr; 44% female; score ≥4 (n = 517) referred to rheumatologist: PsA (n = 175) | • GEPARD | • CASPAR criteria | • Rate of CASPAR fulfilment criteria (score ≥3) in patients with GEPARD score ≥4) |
| 19 | **Haroom_2013** [10], cross-sectional, Ireland | • Group 1: n = 100. Pso whitout PsA (29% PsA), mean age 52.3 yr; 64% male <br> • Group 2: n = 100. Pso with PsA, mean age = 49.6 yr; 49% male | • PASE <br> • PEST <br> • ToPAS | • CASPAR criteria | • Se; Sp <br> • PV+; PV- |
| 20 | **Husni_2007** [13], validation study, USA | n = 69; mean age = 51 yr; female 51% <br> 17/69 (25%) were diagnosed with PsA <br> 24/69 (25%) were diagnosed with OA | • PASE | • Moll & Wright criteria | • Se; Sp <br> • AUC |
| 21 | **Husni_2014** [66], PRISTINE trial | Clinical trial <br> • ETN 50 mg QW: n = 137; mean age 44 yr; male 44% <br> • ETN 50 mg BIW: n = 136; mean age 44 yr; male 65% | • PASE utility in clinical trials (before and after TNF treatment) | • No information about PsA diagnosis | • Baseline: AUC; Sp/Se <br> • Sensitivity to improvement: patients with PASE ≥47 at baseline and week 12 |
| 22 | **Ibrahim_2009** [14], validation study, UK | n = 93 Pso; 12 PsA (12.9%) <br> Addition of other 21 PsA patients | • PEST | • Diagnosis by clinical grounds | • Se; Sp <br> • AUC |

*(Continued)*

**Table 1.** (Continued)

| # | Study | Population | Intervention/s | Gold standard | Measures |
|---|---|---|---|---|---|
| 23 | **Karreman_2017** [67], cross-sectional, The Netherlands | • Primary care<br>• n = 473; 53 PsA (11.2%), mean age 55.7 yr; male 51%, mean Pso duration = 20.7 yr | • PEST<br>• PASE<br>• EARP | • CASPAR criteria | • Se; Sp<br>• AUC |
| 24 | **Khraishi_2010** [50], validation study, Canada | • Rheumatology and dermatology departments (group 1), primary care (group 2)<br>Group 1: n = 89 (n = 59 PsA; n = 30 Pso); 49% male; mean age 52 yr<br>Group 2: n n = 54 (n = 42 PsA; n = 12 Pso); 46% male; mean age 52 yr | • PASQ | • Diagnosis by rheumatologist / CASPAR criteria | • Se; Sp<br>• LR+; LR-<br>• AUC |
| 25 | **Khraishi_2011** [68], validation study, Canada | • Cohort of patients with suspected of early PsA<br>n = 54: 42 with early PsA + 12 without PsA, mean Pso duration 5.2 yr | • ePASQ (PASQ electronic version) | • CASPAR criteria | • Se; Sp<br>• LR+; LR-<br>• AUC |
| 26 | **Leijten_2018** [69], cross-sectional, The Neherlands | • Psoriasis clinic in hospital setting<br>n = 86 Pso; 18 (21%) PsA | • PEST (cut-off 2 and 3) | • CASPAR criteria | • Se; Sp<br>• PV+; PV- |
| 27 | **López-Estebaranz_2015** [70], cross-sectional, Spain | • 40 hospitals. n = 375 Pso; 86 (22.9%) PsA, mean age = 47.4; male 57%; plaque Pso 85% | • PASE | • CASPAR criteria | • Se; Sp<br>• AUC |
| 28 | **Maejima_2016** [71], cross-cultural, Japan | • n = 90 Pso; 19 (21.1%) PsA, 78.8% Pso vulgaris, mean age 57.5 yr (Pso) vs 42.6 (PsA) | • J-EARP | • CASPAR criteria | • Se; Sp<br>• AUC |
| 29 | **Martire_2019** [72], cross-cultural validation, Argentina | • n = 83; 66% women; mean age = 50.7 yr<br>Pso and PsA | • GEPARDa (Spanish version; Argentina) | • CASPAR criteria | • Se; Sp<br>• AUC |
| 30 | **Mazzotti_2019** [73], cross-cultural validation, Brazil | • Tertiary hospital;<br>• n = 124; Pso; 26 (21%) PsA; 50% women; mean age = 51.9 yr, 86% plaque Pso | • PEST-bp (Brazilian Portuguese version) | • CASPAR criteria | • Se; Sp<br>• PV+; PV-<br>• AUC |
| 31 | **Mease_2014** [74], cross-sectional, international, PREPARE study | • 34 dermatology hospitals (Europe & USA)<br>n = 940 pso; 285 (30%) PsA; mean age 49.9 yr; Pso duration = 19.9 yr | Randomization screening with:<br>• PASQ (n = 341)<br>• PEST (n = 332)<br>• ToPAS (n = 340) | • Diagnosis by rheumatologist | • Se; Sp<br>• PV+; PV-<br>• AUC |
| 32 | **Mishra_2017** [75], cross-sectional, India, COMPAQ study | • n = 302; 45 (14.9%) PsA; mean age 40.2 yr; sex (male:female) 2.1:1 | • ToPAS<br>• PASE<br>• PEST<br>• EARP | • CASPAR criteria | • Se; Sp<br>• AUC |
| 33 | **Oyur_2014** [51], cross-cultural validation, Turkey | n = 113 Pso vulgaris; 13 (11.5%) PsA | • PASE | • Moll & Wright criteria | • Se; Sp<br>• AUC |
| 34 | **Piaserico_2016** [76], cross-cultural validation, Italy | • n = 298 plaque Pso; 28% PsA; 56.4% male | • iPASE | • CASPAR criteria | • Se; Sp<br>• AUC |
| 35 | **Salaffi_2018** [77], validation study, Italy | • Multicentric cohort<br>Exclusion: rheumatic diseases mimicking PsA. n = 202 Pso; 62 (30.7%) PsA; mean age 49 yr; 57.9% women | • SiPAS | • CASPAR criteria | • Se; Sp<br>• LR+ |
| 36 | **Tinazzi_2012** [16], validation study, Italy | Derma-rheuma early Pso clinic<br>Exclusion of rheumatic diseases<br>n = 228 Pso; 61(26.7%) PsA | • EARP<br>• PASE | • CASPAR criteria | • PCA: underlying components<br>• Internal consistency<br>• AUC |
| 37 | **Tom_2015** [78], validation study, Canada | Hospital setting<br>n = 556: 131 PsA + 131 Pso & controls<br>Pso (n = 131); PsA (n = 336); controls (n = 89) | • ToPAS 2 (ToPAS2_cap, ToPAS2_uncap) | • CASPAR criteria | • Se; Sp<br>• AUC |

(*Continued*)

**Table 1.** (Continued)

| # | Study | Population | Intervention/s | Gold standard | Measures |
|---|-------|-----------|----------------|---------------|----------|
| 38 | **Urbancek_2016** [79], observational prospective, Slovakia | Outpatient setting<br>n = 831 Pso; 177 PsA (21.8%), plaque Pso 76.9% | • ToPAS<br>• PASE | • CASPAR criteria | • Se; Sp |
| 39 | **Vidal_2016** [80], cross-sectional, Spain | Hospital setting<br>n = 96 Pso patients; 3 (0.03%) PsA 10.2% synovitis, 6.8% enthesitis and tenosynovitis | • PASE<br>• EARP | • Ecographic enthesitis, synovitis, tenosynovitis | • Se; Sp<br>• PV<br>• LR<br>• AUC<br>• Youden index |
| 40 | **Walsh_2013** [81], cross-sectional, USA | Registry-based population<br>n = 189 Pso with ME complaints; 64% PsA | • PEST<br>• PASE<br>• ToPAS | • CASPAR criteria | • Se; Sp |
| 41 | **You_2015** [52], cross-cultural validation, Korea | Dermatology outpatient clinic<br>n = 148 Pso; 18 (12.2%) PsA | • PASE | • CASPAR criteria | • Se; Sp<br>• AUC |

**Abbreviations**: Pso = psoriasis; yr = year; PURE = Psoriatic arthritis UnclutteRed screening Evaluation; PEST = Psoriasis Epidemiology Screening Tool; ToPAS = Toronto Psoriatic Arthritis Screen; PASE = Psoriatic Arthritis Screening Evaluation; EARP = Early Arthritis for Psoriatic patients screening questionnaire; Se = sensitivity; Sp = specificity; PV+ = Positive Predictive Value; PV- = Negative Predictive Value; AUC = area under the curve; CART = classification and regression tree; KR20 = Kunder-Richardson; OA = osteoarthritis; CEPPA = Center of Excellence for Psoriasis and Psoriatic Arthritis; GEPARD = German Psoriasis Arthritis Diagnosis; OA = osteoarthritis; ETN = etanercept; QW = once weekly; BIW = twice weekly; PAQ = Psoriasis Assessment Questionnaire; ePASQ = electronic Psoriasis and Arthritis Screening Questionnaire; SiPAS = Simple Psoriatic Arthritis Screening; ME = musculoskeletal.

were generally poorly described, some patients with psoriasis were on synthetic conventional and biological disease-modifying antirheumatic drugs (DMARDs). However, many of the included studies lacked a clear description of the underlying disease (see Tables 1 and 2). It was not always possible to ascertain the percentage of patients with mild, moderate, or severe disease. In addition, very few articles included healthy subjects or patients with other diseases, who were usually included in a differential diagnosis or cases of difficult diagnosis.

Although most of the analyzed studies included questionnaire-based tools (see below), other tools were also employed, such as the PsA-Disk (a novel 16-item visual instrument) [47] and a combination of serum biomarkers [48].

The following questionnaire-based tools were included in the present SLR: PEST [10, 14, 49, 54–57, 67, 69, 73–75, 81], PASE [10, 13, 16, 49, 51, 52, 55, 56, 59, 60, 63, 66, 67, 70, 75, 79–81], EARP [16, 49, 54, 67, 71, 75, 80], ToPAS [10, 15, 49, 53, 55, 56, 62, 74, 75, 79], ToPAS 2 [61, 78], SiPAT (45) [54], PURE-4 [17], CONTEST [49, 56–58], GEPARD [65, 72], PASQ [50, 74], ePASQ [68], and SiPAS [77]. The number of items included in these questionnaires varied from 4 [17] to 15 [13], and the time to completion varied from less than five minutes to ten minutes (Table 3).

Most studies focused on assessing joint and soft-tissue involvement in PsA (Table 4), all included questions related to joint inflammation, and many included questions about joint pain, dactylitis, and spinal involvement. Four of the questionnaires asked about a prior diagnosis of PsA. The only tools that included psoriatic skin involvement were ToPAS and ToPAS 2. Although most tools shared the same domains, the formulation of questions was generally very different. For example, in questions about dactylitis, PASE and EARP enquired whether the patient has swollen "sausage" fingers, which can facilitate understanding. PEST asked, "Have you ever had heel pain?" but EARP asked, "Has your Achilles tendon become swollen?" Some questionnaires, such as ToPAS and PEST, included images (e.g., of skin, nails) and/or homunculi to aid completion of the questionnaires. Some of the included studies also compared the diagnostic performance of various questionnaires in the same population [10, 16, 49, 54–58, 67, 74, 75, 80].

**Table 2. Main results of included studies.**

| # | Study | Results |
|---|-------|---------|
| 1 | Audureau_2018 [17] | • n = 21 (15.3%) PsA<br>• PURE-4<br>• Se = 85.7% (95% CI 63.7%-98.9%)<br>• Sp = 83.6% (95% CI 75.6%-89.8%)<br>• PV+ = 48.7% (95% CI 31.9%-65.6%)<br>• PV- = 97.0% (95% CI 91.5%-99.4%)<br>• AUC = 87.6% |
| 2 | Chandran_2011 [53] | • n = 169 (24.5%) PsA<br>• ToPAS<br>• Se = 86,8%<br>• Sp = 93.1%<br>• PV+ = 83%<br>• PV- = 94.8%<br>• AUC = 95% |
| 3 | Chimenti_2019 [47] | • PsA-DISK<br>• α Cronbach = 0.90<br>• ICC = 0.98<br>• kappa PsA-DISK vs PEST = 0.46<br>• Construct: PsA + patients: median PsA-Disk score (IQR) = 71 (50–96); PsA-patients: median PsA-Disk score (IQR) = 50 (20–90); p < 0.001 |
| 4 | Chiowchanwisawakit_2016 [54] | • EARP: Se = 83%; Sp = 79.3%; LR- = 0.21; AUC = 90%<br>• PEST: Se = 72%; Sp = 89.7%; LR- = 0.31; AUC = 85%<br>• SiPAT: Se = 91%; Sp = 69%; LR- = 0.13; AUC = 89% |
| 5 | Coates_2013 [55] | • PASE: Se = 74.5%; Sp = 38.5%; AUC = 0.594<br>• PEST: Se = 76.6%; Sp = 37.2%; AUC = 0.610<br>• ToPAS: Se = 76.6%; Sp = 29.7%; AUC = 0.554 |
| 6 | Coates_2014 [58] | AUC (95%CI); (p value). Cutoff and Se<br>• CONTEST: 0.69 (0.57–0.81) (0.01). Cut-off 4: Se = 86%<br>• CONTESTw: 0.74 (0.63–0.85) (0.001). Cut-off 8: Se = 86%<br>• CONTESTjt: 0.70 (0.58–0.72) (0.006). Cut-off 5: Se = 86%<br>• CONTESTtree: 0.59 (0.46–0.73) (0.20)<br>• PEST: 0.61 (0-52-0.70) (0.02)<br>• PASE: 0.59 (0.51–0.68) (0.05)<br>• ToPAS:0.55 (0.46–0.65) (0.27) |
| 7 | Coates_2016 [56] | • CONTEST ≥3: AUC = 0,694; Se = 76%; Sp = 56%<br>• CONTESTjt ≥4; AUC = 0.704; Se = 71%; Sp = 63%<br>• PEST≥2; AUC = 0.652; Se = 82%; Sp = 0.45 |
| 8 | Coates_2018 [57] | • CONTEST: AUC = 0,655; Se = 53%; Sp = 71%<br>• PEST: AUC = 0,723; Se = 60%; Sp = 76% |
| 9 | Costa_2018 [59] | • PASE Portuguese version<br>Cutoff 38: Se = 79%; Sp = 88%; AUC = 0.908<br>α Cronbach = 0.93<br>ICC = 0.97<br>FA: 2 factors with 63% and 7% of explained variance<br>Known groups: median and 25th-75th percentiles significantly higher for PsA vs Pso without PsA |
| 10 | Cretu_2018 [48] | • ITGβ5, M2BP and CRP are independently associated with PsA vs Pso. ROC and AUC = 0.85<br>• Very weak correlations between biomarkers. The largest is ITGb5 and M2BP (r = 0.24) |
| 11 | Dominguez_2009 [60] | • PASE (cut-off ≥ 44): Se = 76%; Sp = 76%<br>• Test-retest: ρ Pearson = 0.35–0.80; ICC = 0.90<br>• Sensitivity to change: Δ after treatment<br>• PsA group: Δ PASE before-after treatment p = 0.034<br>• Non PsA group: Δ PASE before-after treatment p = 0.181 |
| 12 | Duruoz_2018 [61] | • ToPAS 2 (Turkish version)<br>• Se = 95.8%<br>• Sp = 98%<br>• AUC = 0.99 |

*(Continued)*

**Table 2.** (Continued)

| # | Study | Results |
|---|-------|---------|
| 13 | Fernández_Ávila_2017 [62] | • ToPAS (Spanish version; Colombia)<br>• Se = 75%<br>• Sp = 92%<br>• PV+ = 82%<br>• PV- = 88%<br>• FA: 2 factors: dermatologic and rheumatic<br>• Internal consistency: KR20 = 0.73<br>• Test-retest: ρ Pearson = 0.94 |
| 14 | Ferreyra Garrott_2013 [63] | • PASE (Spanish version; Argentina). Cutoff ≥34<br>• Se = 76% (55%-90%)<br>• Sp = 74% (64%-83%)<br>• AUC = 0.79 (0.69–0.89) |
| 15 | Garg_2015 [64] | • CEPPA<br>• Se = 86.9%<br>• Sp = 71.3%<br>• PV+ = 53%<br>• PV- = 93.6%<br>• AUC = 0.87<br>• LR: 1.6–3.7 |
| 16 | Gladman_2009 [15] | • ToPAS: Stepwise regression for discriminative questions<br>• Cut-off 7.71: Se = 94.2%; Sp = 91.5%; PV+ = 81.6;PV- = 97.5; AUC: 0.97<br>• logistic regression based on 3 domains (skin, joint, nails)<br>• Cut-off 7.76: Se = 90.8%; Sp = 90.5%; PV+ = 78.9;PV- = 96.2; AUC: 0.95<br>• Simplified scoring for the ToPAS<br>• Cut-off 8: Se = 86.8%; Sp = 93.1%; PV+ = 83.0;PV- = 94.8; AUC: 0.95 |
| 17 | Haddad_2019 [49] | • PASE: Se = 57.9%; Sp = 93%; PV+ = 89.9;PV- = 67.4; AUC: 0.86<br>• ToPAS: Se = 60.0%; Sp = 93.2%; PV+ = 84.4;PV- = 79.1; AUC: 0.88<br>• PEST: Se = 79.4%; Sp = 94.9%; PV+ = 94.4;PV- = 80.9; AUC: 0.92<br>• EARP: Se = 78.0%; Sp = 91.8%; PV+ = 91.3;PV- = 79.6; AUC: 0.89<br>• CONTEST: Se = 70.0%; Sp = 91.0%; PV+ = 89.3;PV- = 74.0; AUC: 0.87 |
| 18 | Härle_2017 [65] | • University hospital: 43.7% (104/238) GEPARD + and CASPAR+<br>• Doctor's office: 25.8% (68/264) GEPARD + and CASPAR+<br>• Unknown setting: 20.0% (3/15) GEPARD + and CASPAR+ |
| 19 | Haroom_2013 [10] | • Group 1<br>PEST: Se = 27.5%; Sp = 98%; PV+ = 88; PV- = 76<br>PASE: Se = 24%; Sp = 94%; PV+ = 63; PV- = 75<br>ToPAS: Se = 41%; Sp = 90%; PV+ = 63; PV- = 82<br>Group 2:<br>Se of PEST; PASE; and ToPAS = 86%; 62%; 83% |
| 20 | Husni_2007 [13] | • PASE:<br>• Se = 82% (57%-96%)<br>• Sp = 73% (59%-84%)<br>• AUC = 0.84 |
| 21 | Husni_2014 [66] | • PASE<br>• Baseline: AUC = 0.84; Sp/Se = 79.5%<br>• Subjects with PASE ≥47 at baseline and week 12 (%)<br>• ETN 50 QW: 27% - 11%<br>• ETN 500 BIW: 14% - 12% |
| 22 | Ibrahim_2009 [14] | • Alenius PAQ: Se = 63%; Sp = 72%; AUC = 0.76 (0.69–0.85)<br>• PEST: Se = 92%; Sp = 78%; AUC = 0.91 (0.86–0.97) |
| 23 | Karreman_2017 [67] | • PEST: Se = 68% (IC 95% 54%-80%); Sp = 71% (IC 95% 67%-76%); AUC = 0.71<br>• PASE: Se = 59% (IC 95% 44%-72%); Sp = 66% (IC 95% 61%-71%); AUC = 0.64<br>• PASE (cut-off ≥ 44): Se = 66% (IC 95% 52%-79%); Sp = 57% (IC 95% 52%-62%); AUC = 0.68<br>• EARP: Se = 87% (IC 95% 75%-95%); Sp = 34% (IC 95% 30%-39%); AUC = 0.68 |

(*Continued*)

**Table 2.** (Continued)

| # | Study | Results |
|---|---|---|
| 24 | Khraishi_2010 [50] | • PASQ<br>• Group 1<br>• Se = 86.2% (IC 95% 73%-94%); Sp = 88.8% (IC 95% 73%-96%); LR+ = 7.76; LR- = 0.15<br>• AUC = 0.913<br>• Group 2<br>• Se = 70.7% (IC 95% 54.5–83.9); Sp = 81.8% (IC 95% 48.2–97.7); LR+ = 3.89; LR- = 0.36<br>• AUC = 0.881 |
| 25 | Khraishi_2011 [68] | • ePASQ<br>• Se = 97.6%; Sp = 75.0%; LR+ = 3.90; LR- = 0.032<br>• AUC = 0.876 |
| 26 | Leijten_2018 [69] | • PEST: Se = 56%; Sp = 85%; PV+ = 50; PV- = 88<br>• PEST (cut-off ≥2): Se = 89%; Sp = 63%; PV+ = 39; PV- = 96 |
| 27 | López-Estebaranz_2015 [70] | • PASE (cut-off ≥44): Se = 51% (40–62); Sp = 78% (73–83); AUC = 0.707 |
| 28 | Maejima_2016 [71] | • J-EARP: AUC = 0.99; Se = 97.2%; Sp = 97.2% |
| 29 | Martire_2019 [72] | • GEPARDa (cut-off ≥6): Se = 88.6%; Sp = 89.7%; LR+ = 8.6; LR- = 0.12; AUC = 0.955 (0.91–0.99) |
| 30 | Mazzotti_2019 [73] | • PEST-bp: Se = 84.6%; Sp = 63.3%; PV+ = 37.9; PV- = 93.9; AUC = 0.81 (0.73–0.88) |
| 31 | Mease_2014 [74] | • PASQ: Se = 67%; Sp = 64%; PV+ = 43; PV- = 83<br>• PEST: Se = 84%; Sp = 75%; PV+ = 60; PV- = 91<br>• ToPAS: Se = 77%; Sp = 72%; PV+ = 54; PV- = 88 |
| 32 | Mishra_2017 [75] | • EARP: Se = 91%; Sp = 88%; AUC = 0.95<br>• PASE (cut-of ≥44): Se = 80%; Sp = 95%; AUC = 0.93<br>• PEST: Se = 53%; Sp = 95%; AUC = 0.92<br>• ToPAS: Se = 44%; Sp = 97%; AUC = 0.93 |
| 33 | Oyur_2014 [51] | • PASE (cut-off ≥44): Se = 62%; Sp = 76% |
| 34 | Piaserico_2016 [76] | • iPASE (cut-off ≥48): Se = 73.2%; Sp = 76.1%; AUC = 0.821 |
| 35 | Salaffi_2018 [77] | • SiPAS: Se = 79%; Sp = 87%; LR+ = 6.14 |
| 36 | Tinazzi_2012 [16] | • PCA: exclusion of 4 items (EARP of 10 items)<br>α Cronbach = 0.83<br>• AUC PASE (cut-off ≥44) vs EARP: 0.895 vs 0.906<br>• EARP: Se = 85.2%; Sp = 91.6%<br>• PASE (cut-off ≥44): Se = 90.7%; Sp = 67.2% |
| 37 | Tom_2015 [78] | • ToPAS 2_cap (cut-off ≥8): Se = 87.2%; Sp = 82.7%; AUC = 0.910 |
| 38 | Urbancek_2016 [79] | • ToPAS vs PASE<br>• Se: 72.6% vs 58.9%<br>• Sp: 81.3% vs 80.5% |
| 39 | Vidal_2016 [80] | • Tenosynovitis<br>• PASE: Se = 66.7%; Sp = 63.4%; PV- = 93.3%; AUC = 0.65<br>• EARP: Se = 100%; Sp = 48.8%; PV- = 100%; LR+ = 1.81; AUC = 0.74<br>• Synovitis<br>• PASE: PV+ = 12.3%; PV- = 93.5%; AUC = close to 0.50<br>• EARP: Se = 66.7%; Sp = 60.8%; PV- = 94.1%; LR+ = 1.65; AUC = 0.66<br>• Enthesitis<br>• PASE: Se = 100%; Sp = 19.5%; PV+ = 8.3%; PV- = 100%; AUC = 0.66<br>• EARP: PV+ = 8.3%; PV- = 100%; AUC = 0.60<br>• PsA<br>• PASE: Se = 100%; Sp = 66%; PV+ = 9%; PV- = 100%; AUC = 0.83<br>• EARP: not relevant for the PsA diagnosis |
| 40 | Walsh_2013 [81] | • PEST: Se = 85%; Sp = 45%<br>• ToPAS: Se = 75%; Sp = 55%<br>• PASE: Se = 68%; Sp = 50%<br>• PASE (≥44): Se = 78%; Sp = 40% |
| 41 | You_2015 [52] | • Korean PASE (cut-off ≥37): Se = 77.8%; Sp = 82.3%; PV+ = 37.8; PV- = 96.4; AUC = 0.82 (0.72–0.92) |

**Abbreviations**: CI = confidence interval; Se = sensitivity; Sp = specificity; PPV = Positive Predictive Value; NPV = Negative Predictive Value; AUC = area under the curve; FA: factor analysis; KR = Kunder-Richardson; ePASQ = electronic Psoriasis and Arthritis Screening Questionnaire; SiPAS = Simple Psoriatic Arthritis Screening; CEPPA = Center of Excellence for Psoriasis and Psoriatic Arthritis; GEPARD = GErman Psoriasis ARthritis Diagnostic questionnaire

**Table 3. Psychometric characteristics of the included questionnaires.**

| | PEST | PASE | EARP | ToPAS | ToPAS 2 | SiPAT | PURE-4 | CONTEST | CEPPA | GEPARD | PASQ | SiPAS |
|---|---|---|---|---|---|---|---|---|---|---|---|---|
| Validation study | Ibrahim_2009 [14] | Husni_2007 [13] | Tinazzi_2012 [16] | Gladman_2009 [15] | Tom_2015 [78] | Chiowchanwisawakit_2016 [54] | Audureau_2018 [17] | Coates_2014 [58] | Garg_2015 [64] | Härle_2010 [82] | Khraishi_2010 [50] | Salaffi_2018 [77] |
| N° of items | 5 | 15 | 10 | 12 | 13 | 3 | 4 | 8 | 5 | 13 | 10 | 5 |
| Score | 0–5 | 0–75 | 0–10 | 0–12 | 0–13 | 0–3 | 0–4 | 0–8 | 0–5 | 0–13 | 0–15 | 0–5 |
| Cut-off | ≥3 | ≥47 | ≥3 | ≥8 | ≥8 | ≥1 | ≥1 | ≥4 | ≥3 | ≥4 | ≥9 | ≥3 |
| Completion time by patient | NR | 5–6 minutes | < 5 minutes -10 minutes | NR | < 5 minutes -10 minutes | NR | NR | NR | NR | NR | 4 minutes | NR |

**Abbreviations**: NR = non reported

**Table 4. Domains in the PsA screening questionnaires.**

| Domain | PEST | PASE | EARP | ToPAS | ToPAS 2 | SiPAT | PURE-4 | CONTEST | CEPPA | GEPARD | PASQ | SiPAS | Total |
|---|---|---|---|---|---|---|---|---|---|---|---|---|---|
| Joint swelling | X | X | X | X | X | X | X | X | X | X | X | X | 12 |
| Joint pain | | X | X | X | X | | X | | X | X | | X | 8 |
| Bilateral buttocks pain | | | | | | | X | | | X | | | 2 |
| Heel pain | X | | | | | X | X | X | | | | X | 5 |
| Heel swelling | | | X | | | | | | | | | | 1 |
| Morning stiffness | | X | | | | | | | X | X | X | X | 5 |
| Nail involvement | X | | | X | X | | | X | X | | X | X | 7 |
| Dactylitis | X | X | X | X | X | | X | X | | X | | X | 9 |
| Spinal pain | | X | X | X | X | X | | X | | X | X | X | 9 |
| Hot joints | | X | | | | | | X | | | | | 2 |
| Alternating joint pain | | X | | | | | | | | | | | 1 |
| Diagnosis of arthritis | X | | | | | | | | | X | X | | 3 |
| Hand X-ray | | | | | | | | | X | | | | 1 |
| Diagnosis of PsA | | | | X | X | | | | X | | | X | 4 |
| Diagnosis of arthritis other than PsA | | | | X | X | | | | | | | | 2 |
| Age < 50 years | | | | | | | X | | | | | | 1 |
| Work disability | | X | | | | | | | | | | | 1 |
| Self-care impairment | | X | | | | | | | | | | | 1 |
| Problems with wearing rings/watches | | X | | | | | | | | | | | 1 |
| Problems getting in/out of the car | | X | | | | | | | | | | | 1 |
| I'm not as active as I used to be | | X | | | | | | | | | | | 1 |
| The morning is the worst time of day | | X | | | | | | | | | X | | 2 |
| It takes me a few minutes to get in tune any time of day | | X | | | | | | | | | | | 1 |
| Treatment for joint pain | | | X | | | | | | | X | | | 2 |
| Skin involvement | | | | X | X | | | | | | | | 2 |
| You suspect you have an arthropathy | | | | | | | | | | X | X | | 2 |
| Doctor's visit for joint pain | | | | X | X | | | | | X | | | 3 |
| Family history of psoriasis | | | | | X | | | | | | X | | 2 |

**Abreviations**: PsA = psoriatic arthritis

The gold standard for PsA diagnosis also varied. The CASPAR criteria were widely used [10, 15–17, 47–50, 52, 54, 55, 58, 59, 61–63, 65, 67–73, 75, 77–79], but the Moll and Wright criteria [13, 51, 60], rheumatologist's criteria [14, 50, 53, 56, 57, 64, 74], and others were all also employed [80].

Diagnostic performance was analyzed using different outcomes, the most frequent of which were sensitivity, specificity, and AUC. Although different test cut-off points were sometimes analyzed, especially with PASE [51, 52, 70, 76], a single cut-off was evaluated most of the time. Diagnostic performance was generally good/high but very variable. This was expected, and it is probably related to the type of population included and their recruitment.

The AUC of screening tools was higher than 80% in the majority of studies. This was not the case for the following individual studies, which are grouped by PsA screening tool: PASE [55, 58, 67, 80], PEST [55, 58, 59, 67], CONTEST [59], ToPAS [55, 58], PASQ [14], and EARP [67, 80].

Although sensitivities and specificities were generally high, we found a great variability in the results among studies. The data of sensibility were as follows: PEST ranged from 27.5%

[10] to 92% [14], PASE ranged from 24% [10] to 100% [80], EARP ranged from 78% [49] to 100% [80], ToPAS ranged from 41% [10] to 95.8% [61], ToPAS 2 ranged from 87.2% [78] to 95.8% [61], SiPAT was 91% [54], PURE-4 was 85.7% [17], CONTEST ranged from 53% [57] to 86% [58], PASQ ranged from 67% [74] to 86.2% [50], ePASQ was 97.6% [68], and SiPAS was 79% [77]. Specificities were also variable but tended to be even higher than sensitivities: PEST ranged from 37.2% [55] to 98% [10], PASE ranged from 19.5% [80] to 94% [10], EARP ranged from 34% [67] to 97.2% [71], ToPAS ranged from 29.7% [55] to 97% [75], ToPAS 2 ranged from 87.2% [78] to 98% [61], SiPAT was 69% [54], PURE-4 was 83% [17], CONTEST ranged from 71% [57] to 91% [49], PASQ ranged from 64% [74] to 88.8% [50], ePASQ was 75% [68], and SiPAS was 87% [77].

Some of the included studies compared the diagnostic performance of several questionnaire-based tools in the same population [10, 16, 49, 54–58, 67, 74, 75, 80]. Here, we list the screening tools from best to worst, according to the reported AUC: EARP > SiPAT > PEST [54], PEST > PASE > ToPAS [55], COTEST > PEST > PASE > ToPAS [58], PEST > CONTEST [57], PEST > EARP > ToPAS > CONTEST > PASE [49], PEST > EARP > PASE [67], EARP > PASE = ToPAS > PEST [75].

## Telephone interviews

Overall, 68 hospitals in the National Health System were invited to participate, 36 of which accepted the invitation (response rate 53%). These hospitals were distributed throughout the country. Most were teaching hospitals (about half of them are considered referred centers) of different sizes (14% were county hospitals). These hospitals' attended populations varied from <100,000 to >500,000 inhabitants. Thirty hospitals displayed implemented digitalization, such as electronic medical history and citations, while implementation was in progress in a further six hospitals. We found that 20 hospitals (55%) had a multidisciplinary care model for patients with psoriasis and PsA. Patients with suspected PsA were referred to rheumatology from primary care and especially from dermatology departments.

Eleven hospitals had implemented a PsA screening tool or system (30.5%). Some of these (n = 8) used questionnaire-based screening tools in dermatology departments (*e.g.*, PURE-4, PEST, PASE, and modified CASPAR), while others used a system of four predefined questions (n = 2). We also identified one hospital in which primary care doctors that refer patients to that hospital followed specific training in psoriasis and PsA, which included a tele-medical consultation with a dermatologist so as to facilitate early referral of those with suspected PsA. None of the hospitals had analyzed the effectiveness of the screening tools.

Table 5 outlines the barriers and facilitators to implementation of PsA screening tools in clinical practice. In brief, the most reported barriers were lack of time, complexity of the disease, and number and heterogeneity of screening tools. On the other hand, increased collaboration and coordination among physicians, along with the use of new technologies, were identified as the main facilitators to implementing PsA screening tools.

## Nominal group meeting

Finally, a multidisciplinary nominal group meeting was held, in which the results of the SLR and hospital interviews were discussed. First, the experts considered implementing PsA screening tool in primary care and dermatology departments as very useful in order to achieve early diagnosis of PsA. The selection of a screening tool was mainly based on its characteristics and on the features of the Spanish National Health System. Considering most clinics are very busy, the experts agreed that a screening tool with high specificity should be selected, in order to avoid over-referral. However, they agreed that it must also be very simple (i.e., short and

**Table 5. Barriers and facilitators to the implementation of PsA screening tools.**

| Barriers |
| --- |
| Resources: Lack of time (busy clinics), insufficient number of healthcare professionals in departments, limited space for consultations. |
| Disease: Increasing burden of care, complexity of the disease. |
| Screening tools: Lack of training and knowledge about existing tools among physicians, number and heterogeneity of the tools, limitations of the tools' specificity, lack of robust and demonstrated effectiveness of screening tools in real-world settings, patients find self-administered questionnaires difficult. |
| Management: Lack of coordination between departments and care levels (especially rheumatology, dermatology, and primary care), lack of pre-established protocols and processes. |
| **Facilitators** |
| Multidisciplinary care: Collaboration and coordination between dermatology, rheumatology, and primary care; strategies to increase knowledge and improve referral processes, like the implementation of multidisciplinary units, multidisciplinary clinical sessions, and patients education and training. |
| Technology: Online tools/processes to facilitate contact between primary care physicians and dermatologists/rheumatologists, either to set up online consultations or to refer patients, in case of doubt. |
| Management: Increased consultation time, decreased waiting lists. |

**Abbreviations**: PsA = psoriatic arthritis

clear), homogeneous in primary and secondary care, and with good sensitivity. Finally, the experts proposed PURE-4 as the preferential screening tool to be implemented [17]. However, they found no reason to reject other tools, where they are considered appropriate in a given setting, and no reason to replace an existing tool that has already been implemented efficiently. The experts also suggested using the PURE-4 questionnaire at least once per year (but ideally less time), as well as when a patient with psoriasis presents with any suggestive clinical signs or symptoms.

## Discussion

This three-step project was designed to explore PsA screening tools, examine their implementation in clinical practice, and generate related recommendations. There are different PsA screening tools, but their performance is variable depending on the study. Despite its reported effectiveness, their implementation in clinical practice is low. The experts agree that the use of a PsA screening tool can be very useful in clinical practice. The same way they consider that the tool should have a good performance and be simple and clear.

Early referral to rheumatology is central in PsA. It has been observed that up to two thirds of PsA patients present at least one joint erosion at the first visit to a rheumatologist [83]. A significant association between late consulters with peripheral joint erosions and worse Health Assessment Questionnaire scores has also been observed [6]. Moreover, tight control of newly diagnosed PsA patients, including review every four weeks and escalation of treatment if minimal disease activity criteria are not met, showed significant improvement in joint outcomes [84].

We first performed an SLR to assess the characteristics and quality of existing PsA screening tools, regardless of their characteristics. The SLR identified different questionnaire-based screening tools [13–17, 54, 58, 64, 78], a visual instrument named PsA-Disk [47], and a combination of serum biomarkers [48]. Overall, diagnostic performance across these tools was good. However, the results were very variable. This may be partly explained by the heterogeneity of the studies' designs and their included populations (characteristics and recruitment). Therefore, we must be very careful when interpreting the findings of these PsA screening studies. Moreover, when QUADAS-2 was applied, many articles reported a high or unclear risk of

bias, especially with regards to patient selection, which cannot exclude an overestimation of the effect. For example, the study sample was not consecutive or random in many studies, while others lacked a clear description of age ranges, disease severity, type of cutaneous disease or treatments. In addition, only a few of the studies included healthy subjects or patients with other diseases, who should be included for differential diagnosis or in cases where making a diagnosis is difficult. Aligning with our observations, an SLR and meta-analysis that analyzed the diagnostic performance of questionnaire-based PsA screening tools found substantial heterogeneity across studies. Meta-regressions were subsequently conducted, in which age, risk of bias for patient selection, and the screening tool accounted for some of the observed heterogeneity [19].

We also included some articles that compared the diagnostic performance of different questionnaires in the same population [10, 16, 49, 54–58, 67, 74, 75, 80]. Although the results were variable, PEST and EARP presented slightly better performance.

We would also like to highlight that there was great variability in the number of items and domains included in the questionnaires, as well as variability in the way questions were presented. This might have also influenced the results and probably influenced implementation of these tools in daily practice.

Considering all the limitations and considerations that are described above, determining which tool is the most useful for clinical practice is very complicated.

However, it is essential to note the lack of published data from secondary care regarding the level of implementation, feasibility, sensitivity, and specificity of such tools. This prompted us to conduct structured telephone interviews with rheumatology and dermatology physicians, as this enabled us to analyze PsA patients' diagnostic journeys and explore the implementation of PsA screening tools, including their effectiveness (where possible) and barriers and facilitators to their implementation. Only one-third of hospitals reported the use of a PsA screening tool, which included questionnaires, predefined questions, or teleconsultations. All but one of these tools were implemented in secondary care; only one tool was implemented in primary care. Unfortunately, none of the hospitals evaluated the effectiveness of their screening tools in daily practice. However, several barriers to the implementation of PsA screening tools arose, most of which related to a lack of resources (including time or health professionals) and organizational issues. In this sense, more collaboration and coordination between levels of health care and among specialists was found to be essential to ensuring patients are referred in an appropriate and timely fashion, as well as to facilitate the implementation of PsA screening tools. The use of new technologies was also emphasized as a valuable facilitator.

Finally, the experts discussed the results of the SLR and interviews. We would like to comment that this project was performed by a multidisciplinary group of experts, representing all the health professionals involved in the screening of PsA, that have been working closely in different projects in psoriasis and PsA [85–87]. They first addressed the need for implementation of PsA screening tools, in order to facilitate early referral to rheumatology. Although no evidence indicated that a given tool is best, the experts proposed the PURE-4 questionnaire [17, 18] as a promising tool for implementation, considering the features of the National Health System and its clinics. PURE-4 is a recently devised screening tool that only contains four easy-to-collect items. Little or no training is required for its effective implementation and performance. It is therefore suitable for busy clinic, especially in primary care. The experts underlined the simplicity of the questionnaire, its inclusion of some of the most characteristic signs and symptoms of PsA, and the exclusion of others that might lead to over-referral (e.g., spinal pain). Nevertheless, according to the experts, other screening tools are also viable options for implementation in settings that are considered appropriate. Besides, more research is needed with the use of PURE-4.

We would like to comment some of the strengths and limitations of the present study. We have addresses the problem from different perspectives. For this purpose we performed a SLR to assess the performance of PsA screening tools, telephone interviews to analyze the implementation of such tools, barriers and facilitators along with an expert consensus. On the other hand, PsA screening tools available evidence is heterogeneous, and there is a lack of data regarding to their implementation and effectiveness in daily practice. All of this complicates the decision-making. Finally, the selected group of experts tried to be representative including health professionals from rheumatology, dermatology and primary care. However we are aware that the first group was overrepresented and might have influenced some of the discussions.

In summary, early diagnosis and prompt treatment are crucial in PsA [4–6]. This is highlighted in the 2019 update to the EULAR recommendations for the management of PsA with pharmacological therapies [88]. The implementation of PsA screening tools could positively contribute to addressing this situation. In the context of busy clinics, the PURE-4 questionnaire could be a good means of implementing PsA screening.

## Supporting information

**S1 Checklist. PRISMA 2009 checklist.**
(DOC)

**S1 File.**
(DOCX)

## Acknowledgments

**We would like to thank the Cribado working group for their contribution in the project**: José Luis Pinto Tasende, Manel Pujol Busquets, Beatriz Joven, Fernando José Rodríguez Martínez, Eva Galindez, Rubén Queiro, Juanjo Lerma, Teresa Font, Jaime Calvo, Delia Reina, Manuel Moreno, Luis Espadaler, Alex Gómez-Gómez, Mª Dolores López Montilla, Irene Martin, Jose Luis Alvarez Vega, Julio Ramirez, Paulina Cuevas, Javier Manero, Lourdes Mateo, Elena Martínez Castro, Carmen Castro, Natalia Palmou, Carlos Rodríguez Lozano, Agustí Sellas i Fernàndez, Azucena Hernández, Andrea García Valle, Rosa Roselló, Javier Rueda Gotor, Susana Romero, Teresa Clavaguera, Trinidad Pérez Sandoval. The lead author for this group is Ana Urruticoechea-Arana (anboliv@yahoo.es)

## Author Contributions

**Conceptualization:** Ana Urruticoechea-Arana, Estíbaliz Loza, Jordi Gratacós.

**Data curation:** Diego Benavent, Estíbaliz Loza.

**Formal analysis:** Diego Benavent, Estíbaliz Loza.

**Funding acquisition:** Ana Urruticoechea-Arana, Estíbaliz Loza, Jordi Gratacós.

**Investigation:** Diego Benavent, Fernando León, Raquel Almodovar, Isabel Belinchón, Pablo de la Cueva, Cristina Fernández-Carballido.

**Methodology:** Ana Urruticoechea-Arana, Jordi Gratacós.

**Project administration:** Estíbaliz Loza.

**Resources:** Estíbaliz Loza.

**Software:** Estíbaliz Loza.

**Supervision:** Ana Urruticoechea-Arana, Estíbaliz Loza, Jordi Gratacós.

**Validation:** Ana Urruticoechea-Arana, Fernando León, Raquel Almodovar, Isabel Belinchón, Pablo de la Cueva, Cristina Fernández-Carballido, Estíbaliz Loza, Jordi Gratacós.

**Visualization:** Ana Urruticoechea-Arana, Isabel Belinchón, Pablo de la Cueva, Cristina Fernández-Carballido, Jordi Gratacós.

**Writing – original draft:** Ana Urruticoechea-Arana, Estíbaliz Loza.

**Writing – review & editing:** Ana Urruticoechea-Arana, Diego Benavent, Fernando León, Raquel Almodovar, Isabel Belinchón, Pablo de la Cueva, Cristina Fernández-Carballido, Estíbaliz Loza, Jordi Gratacós.

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
