## [Decision Letter · Decision Letter 0]

9 Dec 2020

PONE-D-20-25482

Psoriatic arthritis screening: A systematic literature review and experts’ recommendations

PLOS ONE

Dear Dr. Jordi Gratacós,

Thank you for submitting your manuscript to PLOS ONE. After careful consideration, we feel that it has merit but does not fully meet PLOS ONE’s publication criteria as it currently stands. Therefore, we invite you to submit a revised version of the manuscript that addresses the points raised during the review process.

I agree with the reviewers' comments, recommending minor revisions to the manuscript. Therefore, I suggest to submit  a revised version of the paper and point by point response to the reviewers.

Please submit your revised manuscript in 10 days. If you will need more time than this to complete your revisions, please reply to this message or contact the journal office at plosone@plos.org. Please include the following items when submitting your revised manuscript:

We look forward to receiving your revised manuscript.

Kind regards,

Emiliano Giardina

Academic Editor

PLOS ONE

2. Please provide additional details regarding participant consent. In the ethics statement in the Methods and online submission information, please ensure that you have specified (1) whether consent was informed and (2) what type you obtained (for instance, written or verbal, and if verbal, how it was documented and witnessed).

3. Please include additional information regarding the interview guide used in the study and ensure that you have provided sufficient details that others could replicate the analyses. For instance, if you developed a guide as part of this study and it is not under a copyright more restrictive than CC-BY, please include a copy, in both the original language and English, as Supporting Information. In addition, please provide further details regarding the development and validation of this tool.

"This project was funded by an independent and unrestricted grant from Pfizer."

"IB acted as a consultant and/or speaker for and/or participated in clinical trials sponsored by companies that manufacture drugs used for the treatment of psoriasis, including Janssen Pharmaceuticals Inc, Almirall SA, Lilly, AbbVie, Novartis, Celgene, Amgen, Leo-Pharma, Pfizer-Wyeth, UCB, and MSD; EL has received research grant from Roche, AbbVie, Novartis, Amgen, Leo-Pharma, Pfizer-Wyeth, UCB, Astellas, BMS, Sanofi and MSD; DB received grants/speaker/research supports from Roche, Novartis and Abbvie. PDC acted as consultant, advisory board member, honorary for speaking and participation in clinical trials with the following pharmaceutical companies:  Abbvie, Almirall, Astellas, Biogen, Boehringer, Celgene, Janssen, LEO Pharma, Lilly, MSD, Novartis, Pfizer, Roche, Sanofi, UCB, not related with the submitted work.FL has received honoraria feom Novartis. The rest of authors refer no conflicts of interest."

7. One of the noted authors is a group or consortium (Cribado working group). In addition to naming the author group, please list the individual authors and affiliations within this group in the acknowledgments section of your manuscript. Please also indicate clearly a lead author for this group along with a contact email address.

8. Please include a separate caption for each figure in your manuscript.

Reviewers' comments:

Reviewer's Responses to Questions

**Comments to the Author**

1. Is the manuscript technically sound, and do the data support the conclusions?

Reviewer #1: Yes

Reviewer #2: Yes

2. Has the statistical analysis been performed appropriately and rigorously? 

Reviewer #1: I Don't Know

Reviewer #2: Yes

3. Have the authors made all data underlying the findings in their manuscript fully available?

Reviewer #1: Yes

Reviewer #2: Yes

4. Is the manuscript presented in an intelligible fashion and written in standard English?

Reviewer #1: Yes

Reviewer #2: Yes

5. Review Comments to the Author

Reviewer #1: The paper is interesting, well written and there is no such paper in the literature, it fit with purpose of the journal and seems of interest for the readers.

I only suggest professional revision by a statistician.

Reviewer #2: Dear authors, you report in an exhausting way the relationship between the implementation of PsA

screening tools and PSA prognosis. Moreover, study design and table are easy to understand, clear

and adequate. I report some notes to improve your manuscript:

1) Implement introduction and relation between pso and psa from:

Caputo V, Strafella C, Termine A, Dattola A, Mazzilli S, Lanna C, Cosio T, Campione E, Novelli G,

Giardina E, Cascella R. Overview of the molecular determinants contributing to the expression of

Psoriasis and Psoriatic Arthritis phenotypes. J Cell Mol Med. 2020 Oct 31. doi:

10.1111/jcmm.15742. Epub ahead of print. PMID: 33128843.

2) Line 151, 298, of, correct

3) Page 20, correct caption

4) Page 25, line 281-290. Did you consider national center for psoriasis? General hospitals’ attended

populations could not reflect PSA center population.

5) Table 5, this is a hotspot, really interesting.

6) Pag 27 line 323-327. Other consideration in timing?

6. PLOS authors have the option to publish the peer review history of their article (what does this mean?). If published, this will include your full peer review and any attached files.

Reviewer #1: No

Reviewer #2: No

---

## [Author Response · Author response to Decision Letter 0]

12 Feb 2021

Manuscript [PONE-D-20-25482] - [EMID:631d5c122f91803b]

Barcelona, December 18th, 2020

Dear editor, 

Thank you very much for the opportunity to revise our manuscript by answering the reviewers´comments. And thank you as well to the reviewers for the improving the readiness of our manuscript with their suggestions. We have replied to each one of them, as you can see below. 

We hope, these responses and changes clarify the study.

Sincerely, 

Jordi Gratacós on behalf of the authors. 

Reviewer #1: The paper is interesting, well written and there is no such paper in the literature, it fit with purpose of the journal and seems of interest for the readers.

I only suggest professional revision by a statistician.

Thank you for your comment. We have consulted a expert statistician and we have made some changes. 

Reviewer #2: Dear authors, you report in an exhausting way the relationship between the implementation of PsA screening tools and PSA prognosis. Moreover, study design and table are easy to understand, clear and adequate. I report some notes to improve your manuscript:

Thank you very much

1) Implement introduction and relation between pso and psa from:

Caputo V, Strafella C, Termine A, Dattola A, Mazzilli S, Lanna C, Cosio T, Campione E, Novelli G, Giardina E, Cascella R. Overview of the molecular determinants contributing to the expression of Psoriasis and Psoriatic Arthritis phenotypes. J Cell Mol Med. 2020 Oct 31. doi: 10.1111/jcmm.15742. Epub ahead of print. PMID: 33128843.

Thank you very much for your recommendation. 

2) Line 151, 298, of, correct

Done. 

3) Page 20, correct caption

Done

4) Page 25, line 281-290. Did you consider national center for psoriasis? General hospitals’ attended populations could not reflect PSA center population.

Thank you for your comment. We have made some changes. 

Most were teaching hospitals (about half of them are considered referred centers) of different sizes (14% were county hospitals)

5) Table 5, this is a hotspot, really interesting.

Thank you for your comment

6) Pag 27 line 323-327. Other consideration in timing?

We have added some information

---

## [Editor Report · Decision Letter 1]

2 Mar 2021

Psoriatic arthritis screening: A systematic literature review and experts’ recommendations

PONE-D-20-25482R1

Dear Dr Gratacós,

We’re pleased to inform you that your manuscript has been judged scientifically suitable for publication and will be formally accepted for publication once it meets all outstanding technical requirements.

Kind regards,

Emiliano Giardina

Academic Editor

PLOS ONE

---

## [Editor Report · Acceptance letter]

4 Mar 2021

PONE-D-20-25482R1 

Psoriatic arthritis screening: A systematic literature review and experts’ recommendations 

Dear Dr. Gratacós:

I'm pleased to inform you that your manuscript has been deemed suitable for publication in PLOS ONE. Congratulations! Your manuscript is now with our production department. 

Kind regards, 

on behalf of

Dr. Emiliano Giardina 

Academic Editor

PLOS ONE